# Field Trial Vaccination against Cowpox in Two Alpaca Herds

**DOI:** 10.3390/v12020234

**Published:** 2020-02-20

**Authors:** Almut Prkno, Donata Hoffmann, Matthias Kaiser, Daniela Goerigk, Martin Pfeffer, Karsten Winter, Thomas W. Vahlenkamp, Martin Beer, Alexander Starke

**Affiliations:** 1Clinic for Ruminants and Swine, Faculty of Veterinary Medicine, University of Leipzig, An den Tierkliniken 11, 04103 Leipzig, Germany; mkaiser@vetmed.uni-leipzig.de (M.K.); alexander.starke@vetmed.uni-leipzig.de (A.S.); 2Institute of Diagnostic Virology, Friedrich-Loeffler-Institut, Federal Research Institute for Animal Health, Südufer 10, 17493 Greifswald-Insel Riems, Germany; Donata.Hoffmann@fli.de (D.H.); Martin.Beer@fli.de (M.B.); 3Veterinary practice Dr. Daniela Goerigk, Naundorfer Str. 9, 04668 Schkortitz, Germany; info@tierarzt-goerigk.de; 4Institute of Animal Hygiene and Veterinary Public Health, Centre for Veterinary Public Health, Faculty of Veterinary Medicine, University of Leipzig, An den Tierkliniken 1, 04103 Leipzig, Germany; pfeffer@vetmed.uni-leipzig.de; 5Institute of Anatomy, Faculty of Medicine, University of Leipzig, Liebigstraße 13, 04103 Leipzig, Germany; kwinter@rz.uni-leipzig.de; 6Institute of Virology, Faculty of Veterinary Medicine, University of Leipzig, An den Tierkliniken 29, 04103 Leipzig, Germany; vahlenkamp@vetmed.uni-leipzig.de

**Keywords:** alpaca, South American camelids, cowpox, prevention, vaccination, MVA

## Abstract

In Europe, cowpox virus (CPXV) infection in South American camelids occurs as a so-called spill-over infection. Although infected animals generally have a mild form of the disease and survive, cases of fatal generalised CPXV infection have also been described. Prevention by prophylactic vaccination is the only way to protect animals from disease. In the present study, modified vaccinia virus Ankara (MVA) vaccine, which has been successfully used in many animal species, was used in a prime-boost vaccination regimen in two alpaca herds with a history of CPXV infection. The focus of the study was the prevention of further clinical cases, and to determine the safety and immunogenicity of the MVA vaccine in alpacas. The MVA vaccine was well tolerated and safe in the 94 animals vaccinated. An indirect immunofluorescence assay (IFA) using MVA as an antigen showed that the seroprevalence of antibody after booster vaccination was 81.3% in herd I and 91.7% in herd II. Detectable antibody titres declined to 15.6% in herd I and 45.8% in herd II over a 12-month period after booster vaccination. Animals could be divided into four groups based on individual antibody titres determined over one year: Group 1 consisted of 19.3% of animals that were seropositive until the end of the trial period; Group 2 consisted of 58.0% of animals that were seropositive after booster vaccination, but seronegative one year later; Group 3 consisted of 14.7% of animals that were not seropositive at any time point; and Group 4 consisted of 7.9% of animals that were seropositive after initial immunisation, seronegative six months later, but seropositive or intermediate in IFA one year after immunisation, likely because of natural exposure. In new-born crias born to MVA-vaccinated mares, specific maternal antibodies were detected in 50.0% of animals up to 14 weeks of age. Our results confirm that MVA vaccination is a feasible tool for the prevention of CPXV disease in alpacas. Long-term studies are needed to verify future vaccination regimen in CPXV affected herds.

## 1. Introduction

Cowpox is a disease caused by the cowpox virus (CPXV), a member of the genus Orthopoxvirus (OPV), family Poxviridae [1]. Being endemic in Eurasia, a number of animal species have been shown to be naturally susceptible to CPXV [1]. In addition, certain non-native animal species, kept in zoos or other animal holdings, are also susceptible to CPXV infections. Examples include African rhinoceroses and elephants [2,3,4,5,6] as well as jaguarundis, giant anteaters, ocelots, Patagonian cavies and New World monkeys [7,8,9,10,11]. CPXV infection in South American camelids (SACs) was first described in 1997 [12], and since then, only a few reports of CPXV infection in SACs have been published [13,14,15,16,17]. CPXV infections in SACs occur sporadically. Reservoir hosts (i.e., circulation of the pathogen within a population) of CPXV are likely wild rodents (bank voles, common voles and striped field mice) and virus transmission through indirect contact is probably the most common route of transmission. Infection of SACs with CPXV leads to two different disease presentations: mild, mostly self-limiting infections with localised skin lesions (pustules and crusts), and generalised, frequently lethal infections with multifocal to diffuse skin lesions (papules, pustules, crusts, ulcers) accompanied by virus replication in other organs. Although llamas and alpacas are an established species of livestock and pets in Europe, there are currently no data on the incidence of CPXV infection in SACs.

Research on antiviral agents, such as cidofovir, brincidofovir (CMX001) and tecovirimat (ST-246), for the treatment of OPV infections is ongoing [18,19,20,21,22]. However, to the authors’ knowledge, there is currently no approved treatment for CPXV infection in animals. Thus, prevention via prophylactic vaccination is the only feasible approach to manage CPXV diseases, especially in valuable and highly susceptible animals (e.g., elephants and other zoo animals). 

It is well established that each species of the genus OPV induces cross-reactive antibodies, which provide immunoprotection against infection with other species of OPV [1]. Therefore, prophylactic vaccination remains an important tool in the prevention of diseases caused by orthopoxviruses. One of the best examples is the World Health Organization’s global campaign to eradicate human smallpox, which is caused by variola virus, using the vaccinia virus vaccine. Different vaccine strains of vaccinia virus were available for this campaign [23]; the strain Elstree was used in Germany [24]. At the end of this campaign (1972–1980), another virus strain called modified vaccinia virus Ankara (MVA) was licensed in Germany as part of the official smallpox vaccination program for humans [25].

MVA is a highly attenuated strain originating from a virulent vaccinia virus strain called chorioallantois vaccinia virus Ankara. It was developed by Mayr and colleagues in the 1960s and 1970s [24,25] through more than 500 passages of chorioallantois vaccinia virus Ankara in chicken embryo fibroblasts. This resulted in six large deletions within the terminal genome fragments leading to an overall reduction of the genome length from 208 to 178 kilo base pairs. By the 530th passage in chicken embryo fibroblasts, the strain was called MVA. Genes that are affected by deletion regulate host interaction and the production of A-type inclusion bodies. Attenuations lead to a greatly restricted host range and the inability of the virus to complete replication cycles in mammalian cells [25,26]. In humans, the MVA vaccine is avirulent, non-contagious and well tolerated, which renders it safe for use in immunosuppressed individuals [27,28]. In 2013, Imvanex (Bavarian Nordic A/S), the only licensed MVA vaccine in Europe, was approved by the European Medicines Agency to vaccinate humans against smallpox infection [29]. In addition, this vaccine was used off-label for pre- and post-exposure prophylaxis in humans with monkeypox virus infection in the United Kingdom [30].

The MVA-based vaccine is also used to protect animals from CPXV-induced disease and has been primarily described in elephants [25,31] but also in mice, domestic cats, rabbits and other zoo animals [9,25,32,33,34]. By German law, this vaccine requires a special exemption issued by the appropriate Ministries of the Federal States.

The use of MVA vaccine in alpacas or llamas has not been described so far. The economic value of these animals and the zoonotic potential of CPXV infection justify determination of whether MVA vaccine is effective for the prevention of CPXV-induced disease in SACs. The present study describes the application of the MVA vaccine in a prime-boost immunisation regimen in two alpaca herds in Germany with a history of CPXV infections [16]. The aim of the present study was to assess the safety of a live MVA vaccine in alpacas and to determine its immunogenicity after initial immunisation and one year later.

## 2. Materials and Methods 

### 2.1. Alpaca Herds

The study included a total of 102 alpacas (35 males, 67 females) that varied in age and originated from two privately owned herds (herd I – Thuringia, *n* = 69; herd II – Saxony Anhalt, *n* = 33) in Germany. The owners of each herd had divided the animals into gender-specific groups, which had no effect on the study. The husbandry systems used were perennial open housing (herd I) and a combination of pasture feeding in the summer and open housing in the winter (herd II). No other farm or companion animals were kept on either of the farms. 

### 2.2. Herd History

In both herds, clusters of CPXV infection occurred and were confirmed by determining CPXV-specific antibody titres and CPXV-specific DNA, as described by [16]. Herd seroprevalences of 16.4% (herd I) and 16.1% (herd II) were determined at that time. To prevent further cases of CPXV infection, MVA vaccination was considered to be the best option. There was a period of 13 months (herd I) and three months (herd II) between the last herd investigation [16] and the start of MVA vaccination. A total of 69 animals (64 adults and 5 crias) of herd I and 33 animals (24 adults and 9 crias) of herd II were used in the present study. This included seven animals of herd I (IDs 101, 103, 138, 139, 145, 165 and 166) and two animals of herd II (IDs 12 and 16) that tested positive for CPXV-specific antibody in dilutions ranging from 1:200 to 1:500 [16]. CPXV-specific antibody titres determined in previous studies were henceforth referred to as ‘pre-existing field titres’ (Appendix A).

### 2.3. MVA Vaccination

A total of 94 of 102 animals (herd I – *n* = 64; herd II – *n* = 30; 8 animals had not been born at that time) were vaccinated with the MVA vaccine ‘MVA F6 LMU SF 12-9′ (>10^7,5^ TCID/mL in chicken embryo fibroblasts) provided by the Institute for Infectious Diseases and Zoonoses, Ludwig Maximilians University München, Munich, Germany (Table 1). A special exemption was obtained from the relevant Ministries of the Federal States (herd I: Thuringian Ministry of Labour, Social Welfare, Health, Woman and Family Affairs - permit-number: 51-2511/90-1-33743/2013, July 15, 2013; herd II: Ministry of Environment, Agriculture and Energy of Saxony-Anhalt – permit-number: 65-42114/1, May 25, 2013) under German law (Animal Health Act Sect. 11 para. 5 Nb 1). Animals were vaccinated subcutaneously on the right side of the neck, cranial to the scapula, and a booster vaccination was administered four weeks later at the same location. A 2.0 mL aliquot of vaccine was used in adult alpacas and 1.0 mL of vaccine was used in crias (up to six months of age). The route of administration and volume of vaccine were chosen according to the vaccine’s batch record and reports in other animal species [24,25].

### 2.4. Sampling

Four weeks after the initial and booster vaccinations, 88 animals underwent clinical examination as described by [16], were checked for vaccine reactions and had a jugular blood sample and a swab of the oral mucous membranes collected as described by ([16], Table 1). Jugular blood samples and an oral swab were again collected six and 12 months after the booster vaccination in all animals. 

To determine the presence of specific serum maternal antibodies (maAbs), a jugular blood sample was collected two to 12 weeks after birth in 14 crias born to dams that had been vaccinated with MVA before or during pregnancy. Crias (*n* = 6) born at the start of the study between time points 1 and 3 were vaccinated twice with MVA vaccine, immediately after initial blood sampling and four weeks later (outside the time points outlined in Table 1). At the time of booster vaccination, possible vaccine reactions were recorded, and a jugular blood sample was collected. These crias were again sampled at time points 4 and 5 (Table 1) together with the adult herd. Crias (*n* = 8) born near the end of the study between time points 4 and 5 were not vaccinated with MVA vaccine because special permission for the vaccine had expired. Five of these crias were sampled for the second time at time point 5 together with the adult herd and the remaining three were sampled for the first time at time point 5.

### 2.5. Diagnostic Methods

To determine the safety of the MVA vaccine in alpacas, a scoring system described by [35] was used to classify adverse reactions. Four weeks after each immunisation, the owners were interviewed regarding score K, which denoted general signs of illness including fever, anorexia and listlessness in the week following immunisation. Score A described abscess formation at the injection site. At the same time, chronic localized dermal swelling was measured using a Vernier calliper (MarCal 16 DN Messschieber mit Skalenanzeige, Mahr GmbH, Göttingen, Germany) and scored based on diameter (0, no reaction; 1, ≤2 cm; 2, >2 cm and <5 cm; 3, ≥5 cm).

An indirect immunofluorescence assay (IFA) using endpoint dilution was carried out to detect specific antibodies in sera of vaccinated alpacas and new-born crias. Briefly, MVA-infected BHK21 cells (Collection of Cell Lines in Veterinary Medicine, Friedrich-Loeffler-Institute, Greifswald-Insel Riems, Germany) and non-infected control cells were incubated with diluted alpaca serum after fixation using a methanol/acetone protocol. After incubation for one hour at room temperature, cells were washed with phosphate-buffered saline, and a FITC-labelled secondary antibody (Bethyl laboratories, Montgomery, Texas, US) was applied for one hour at room temperature. The cells were washed, and immunofluorescence microscopy was used to detect primary antibody binding. Each IFA was validated by the use of reference alpaca sera. In animals with inconclusive antibody results, a second IFA based on CPXV-infected cell cultures was done as previously described [16,36]. Generally, sera scored positive only after viral plaque-associated staining. Sera were deemed “intermediate” when there was a non-plaque associated reaction pattern. Titres of 200 and greater were considered positive (≥1:200), sera non-reactive using a 1:200 dilution scored negative (<1:200) and sera reactive with titres between 0 and 200 were considered intermediate (>0<1:200). To show antibody titres of individual animals over time (P), results for each time point were expressed with symbols as follows: ‘+’, positive, ‘-‘, negative and ‘?’, intermediate. For statistical analysis, intermediate results were assigned a value of zero. Serum samples collected from alpacas in a previous study [16] were re-tested using IFA (MVA) to confirm the pre-existing titres, which were a response to natural infection. Oral mucosal swab specimens collected from the alpacas were used for quantitative real-time polymerase chain reaction for the detection of OPV-specific DNA as described previously [36,37].

### 2.6. Statistical Analysis

Statistical analysis was done with IBM^®^ SPSS^®^ Statistics (version 22; IBM Corp., Armonk, New York, USA) and Mathematica^®^ (Version 11.3, Wolfram Research Inc., Champaign, IL, USA). Descriptive statistics were calculated, bar charts and line charts were generated, and data were tested for normal distribution using the Shapiro–Wilk test. Group comparisons were done using t-tests (normal data) and Mann–Whitney U tests (non-normal data). A Fisher’s exact test was used to analyse data in contingency tables. Significance was set at *p* < 0.05.

## 3. Results

### 3.1. Safety

In herd I, one of 64 animals (1.6%) had a vaccine reaction four weeks after the initial vaccination (time point 2). The vaccine reaction could be seen and palpated as a localised dermal swelling, 3.0 cm in diameter (score 2). There were no vaccine reactions at time point 3, which was four weeks after the booster vaccination.

In herd II, two of 24 animals (8.4%) had a vaccine reaction. The first occurred one day after the initial vaccination (time point 1) and manifested as a localised dermal swelling, 1.0 cm in diameter (score 1). The second was an adverse reaction (score K), which occurred immediately after the initial vaccination and was characterised by anorexia and listlessness for two days. Vaccine reactions did not occur after the booster vaccination. 

### 3.2. Immunogenicity

Serological testing showed antibody titres after the first and second vaccinations in both herds (Figure 1 and Appendix A). 

In herd I, IFA showed detectable antibody titres in 24 of 64 animals (37.5%) four weeks after the initial vaccination (time point 2). Four weeks after the booster vaccination (time point 3), 52 of 64 animals (81.3%) had seroconverted. The number of seropositive animals decreased to 10 of 64 animals (15.6%) six months after the booster vaccination and remained at this level until 12 months after the booster vaccination (Figure 1A). The results of individual animals are shown in Appendix A.

In herd II, IFA showed detectable antibody titres in 11 of 24 animals (45.8%), four weeks after the initial vaccination (time point 2). Four weeks after the booster vaccination (time point 3), 22 of 24 animals (91.7%) had seroconverted. The number of seropositive animals decreased to 11 of 24 animals (45.8%), six months after booster vaccination, and remained constant in 11 animals until 12 months after booster vaccination (Figure 1B).

The distribution of MVA-specific titres and the mean titres for each herd and time point are shown in Figure 1C.

The mean titres were similar in both herds during the study. They increased from time point 2 to time point 3 and then declined from time point 3 to time point 4 and decreased further until time point 5. The mean titre curve of herd I was lower than that of herd II at all time points. Significant differences between the two herds occurred at time point 3 (U_(TP3)_ = 1058.0; p_(TP3)_ = 0.005), time point 4 (U_(TP4)_ = 1006.0; p_(TP4)_ = 0.003) and time point 5 (U_(TP5)_ = 997.5; p_(TP5)_ = 0.004), but not at time point 2 (U_(TP2)_ = 890.5; p_(TP2)_ = 0.193).

Table 2 shows the wide range in antibody titres over time (P) for individual animals. Although there were 18 possible variations (herd I – 15; herd II – 12), these values could be used to divide animals into four groups. Group 1 (P 1) represented the animals (*n* = 17; 19.3%) that had stable positive antibody titres throughout the study. This group included animals with titres of up to 1:8,000, especially early in the study. In addition, there were nine animals (IDs 12, 16, 101, 103, 136, 138, 139, 145 and 166) that had had natural CPXV infection before the start of the study; this was confirmed by determination of either the CPXV-specific antibody titre [16] or the MVA-specific antibody titre (animal ID 136). The remaining eight animals (IDs 1, 2, 3, 15, 17, 20, 25 and 118) had not had natural CPXV infection. Within this group, there were no differences in antibody titre endpoint dilutions between CPXV-naïve and CPXV-exposed animals at any of the four time points (U_(TP2)_ = 45.0, p_(TP2)_ = 0.423; U_(TP3)_ = 46.5, p_(TP3)_ = 0.321; U_(TP4)_ = 43,5, p_(TP4)_ = 0.481; U_(TP5)_ = 53.0, p_(TP5)_ = 0.114).

Group 2 (P 2 – 10) included animals (*n* = 51; 58.0%) that developed positive antibody titres before time point 3. Thereafter, their titres decreased and were negative by time point 5 (except for animal ID 8—P 3). P 2 and P 3 consisted of animals that tested positive at three time points, P 4 and P 5 were animals that tested positive at two time points, and P 6 to P 10 included animals that seroconverted at one time point only. Within group 2, there were individual titres of up to 1:4000. A large number of individual titre changes occurred after P 4 and P 9. Animals of P 4 had seroconverted by time points 2 and 3 with dilutions ranging from 1:200 to 1:500 and 1:200 to 1:2000, respectively. The animals of P 9 had seroconverted only by time point 3, with dilutions ranging from 1:200 to 1:1000. Comparison of the antibody titre endpoint dilutions showed that the animals of group 1 had consistently higher antibody titres than the animals of group 2 at all four time points (U_(TP2)_ = 14.5, p_(TP2)_ < 0.001; U_(TP3)_ = 157.5, p_(TP3)_ < 0.001; U_(TP4)_ = 20.0, p_(TP4)_ < 0.001; U_(TP5)_ = 8.5, p_(TP5)_ < 0.001).

Group 3 (P 11 – 13) comprised animals (*n* = 13; 14.7%) in which seroconversion did not occur at any time point in the study. Group 4 (P 14 – 18) included animals (*n* = 7; 7.9%) that had fluctuations in antibody titres throughout the study. Seroconversion occurred by time point 3 at the latest and dilutions ranged from 1:200 to 1:2000. Titres were negative at time point 4 but were detectable again (positive at 1:200 or intermediate result) at time point 5.

The sera of animals of groups 3 (no seroconversion) and 4 (inconclusive titre changes) underwent additional testing in a second IFA using CPXV as antigen (Table 3). Results showed that four animals in group 3 (IDs 140 and 153 – P 12, IDs 151 and 163 – P 13) were negative in both the IFA MVA and IFA using CPXV as antigen. Of the seven animals in group 4, one (ID 10 – P 15) had inconclusive results in both tests. 

### 3.3. Results of Crias

Of 14 crias (herd I, *n* = 5; herd II, *n* = 9) tested, seven (50.0%; herd I, *n* = 2, herd II, *n* = 5) had positive antibody titres ranging from 1:200 to 1:1000 at the time of initial blood sampling in the first weeks of life, indicating the presence of specific serum maternal antibodies (maAbs) to MVA. The age of crias that tested positive for maAbs ranged from 2.9 to 6.6 weeks (Table 4). Cria no. 172 also tested positive (1:200) at 14.1 weeks of age (time point 5) even though it had not received the MVA vaccination. The crias that tested negative for maAbs to MVA had a mean age of 6.5 ± 3.1 weeks, compared with 4.2 ± 1.3 weeks for crias that had maAbs to MVA; however, the difference was not significant (t(8) = 1.83; p = 0.105). The two groups of crias had the same female-to-male ratios and thus there was no effect of sex (p = 1.000; OR = 1.00; 95%CI (lower) = 0.120; 95%CI (upper) = 8.307). There appeared to be a trend toward a positive association between crias with positive maAb titres born to dams with positive antibody titres, whereas crias with no maAb titre to MVA were born to dams with positive or negative antibody titres. There was no effect of titre status of the dam on the maAb titre status of the cria (p = 0.070; OR = 0.00; 95%CI (lower) = 0.00; 95%CI (upper) = N/A). There was a trend for a higher maAb titre prevalence in crias born to dams with a short interval between foaling and vaccination, but the difference was not significant (p = 0.592; OR = 3.33; 95%CI(lower) = 0.362; 95%CI(upper) = 30.703). Vaccine reactions were not seen in crias (*n* = 6) that were vaccinated with the MVA vaccine after initial blood sampling; these six crias were seronegative at time points 4 and 5. 

### 3.4. Results of Swab Samples

None of the oral swab samples tested positive for OPV-specific DNA in herd I or II at any time points in the study.

## 4. Discussion

Two separate clusters of CPXV infection in two alpaca herds in Germany [14,16] prompted MVA vaccination to prevent further cases of cowpox disease and fatalities. To the authors’ knowledge, this had not yet been described, and therefore provided a unique opportunity to study the safety and effectiveness of the MVA vaccine in alpacas in the field. 

Our results showed that the MVA vaccine was safe and well tolerated in both alpaca herds. Mild self-limiting vaccine reactions occurred after the initial vaccination in only three of 94 animals. No other side effects were observed at any time point.

The virulence of MVA is considerably reduced compared with the field strain of CPXV and therefore a minimal effective virus concentration of 10^7.5^ TCID/mL vaccine, administered twice parenterally, three to four weeks apart, is recommended to achieve protective immunity against cowpox [25]. This same prime-boost vaccination regimen using a dose of 2.0 mL for adults and 1.0 mL for crias was implemented in the present study and led to detectable antibody titres in most of the animals after the first or second vaccination. A noticeable increase in antibody titres after the second vaccination in both herds confirmed the effect of the booster vaccination. Our results are in agreement with those of other studies on cynomolgus monkeys [38], rhesus monkeys [39], domestic cats [33], elephants [25,31] and rabbits [34].

The results of a one-year follow-up evaluation showed changes in individual antibody titres, which were group dependent but similar in both herds. A decrease in antibody titre over the one-year period was common in most of the animals, especially in groups 1 and 2. However, there were marked differences in the speed and degree of the titre decrease. Group 1 animals had stable positive antibody titres, which were only decreased at the last time point. This group consisted of CPXV-naïve and CPXV-exposed animals, which had analogous titre changes, indicating that CPXV-naïve animals had a very good immune response to the MVA vaccine, or, more likely, had had a CPXV infection that was not detected. The CPXV-specific antibody titres of animals that had been exposed to natural CPXV infection increased after MVA vaccine, which had a boosting effect. A similar boosting effect post-infection was described in jaguars, snow leopards, jaguarundis [9] and domestic cats [33].

The animals in group 2 had a decrease in titre or negative titre six months after the second vaccination or at the final end point. There are no long-term studies on the duration of protective immunity after MVA vaccination in animals. Human studies on the safety and immunogenicity of the MVA vaccine [23,29,40] showed that titres decreased by six months and two years after initial immunisation using two doses of vaccine administered four weeks apart. In addition, antibody titres of vaccinia-naïve and vaccinia-experienced subjects differed with significantly higher titres two years after booster vaccination in the latter [23,29]. Similar conclusions can be drawn from our data because MVA vaccination of CPXV-naïve animals of group 2 did not result in antibody titres as high or as long-lasting as those in the CPXV-exposed animals of group 1. Nevertheless, the majority (58%) of vaccinated alpacas had a decrease in or negative antibody titre one year after prime-boost vaccination, and we assumed that this was attributable to an alpaca-specific reaction to the MVA vaccination in CPXV-naïve animals.

Group 3 had four animals that had no antibody production as determined by the IFA. It is possible that these animals had no immunological response to MVA vaccination or that their antibody titres were below the detection limits of our test systems. More sensitive assays and test systems that determine cellular immunity in alpacas are needed to address this issue in future studies.

The fluctuations in antibody titres seen in group 4 were of interest. The increase in titres more than six months after the booster vaccination may have been attributable to the new exposure to CPXV in the field, especially in animal No. 10, which had reproducible results in both tests. CPXV infections in animals often occur in seasonal peaks (winter; late summer/autumn) during the year [16]. Time points 4 (February/May) and 5 (July/November) corresponded to these seasonal peaks, and thus exposure to CPXV in the environment cannot be ruled out.

All oral swab samples collected for the detection of acute CPXV infection tested negative for OPV-specific DNA. Based on the results of studies in other alpaca herds [16], a negative oral swab sample alone does not rule out CPXV infection. Therefore, thorough clinical examination including careful palpation of the skin and fleece, especially the cranial areas, and complete and fastidious sampling of crusts and suspicious lesions of the skin and mucous membranes, is required to confirm acute CPXV infection. Nevertheless, based on our results, we assumed that no acute clinical CPXV infection was present at the times of sampling.

Another aim of this study was to determine whether crias born to vaccinated mares have specific serum maternal antibodies to MVA in the first weeks of life. Our results confirmed that MVA-specific maAbs were detectable in new-born crias up to an age of 14 weeks. We also showed that this status was closely associated with a positive antibody titre in the mare. Maternal immunoglobulins are transferred to crias via the colostrum because the epitheliochorial placenta does not allow passage of immunoglobulins from dam to fetus [41]. This means that a positive maternal titre as well as adequate colostrum intake in the first hours of life are essential for the immunoprotection of crias in the first weeks of life. Based on our results, it is very likely that MVA vaccination of mares provides protection of new-born crias against clinical CPXV infection. Therefore, it is best to vaccinate mares in late pregnancy, at least four weeks before parturition, to ensure high antibody titres. 

In our study, we vaccinated crias with MVA vaccine immediately after initial sampling in the first weeks of life, even though maternal immunoglobulins were present. Because all crias were seronegative six and 12 months after vaccination, we concluded that vaccination of crias with maAb titres is not appropriate. The half-life of maternal immunoglobulins in alpacas and llamas is believed to be 10 to 23 days, with the disappearance of 97% of colostral maternal immunoglobulins by 115 days [41]. It is also well known that maternal antibodies hinder the antigenic response to vaccination [42]. Thus, initial vaccination of crias should be done after the loss of passive immunity afforded by maAbs, which is reported to occur by three-and-a-half months of age at the earliest [41]. 

In summary, the MVA vaccination of alpacas is feasible and well tolerated. It is not known whether the titres measured are protective against natural CPXV infection, because the value of the animals and animal welfare regulations precluded experimental challenge with CPXV. It appears that MVA is as immunogenic as the field virus [43], and we feel that this vaccine would provide adequate immunoprotection because no new cases of fatal CPXV infection were observed in the two herds after immunisation. Although most of the alpacas had a negative titre one year after prime-boost vaccination, it is likely that once an animal has responded to CPXV or MVA, immunological memory would enable protection in the face of CPXV infection. This may have been the situation in animal ID 10, which had a detectable antibody titre six months after the titre had declined to an undetectable level. 

## 5. Conclusions

Alpaca herds are constantly at risk of CPXV infection because of their husbandry systems, which allow contact between the animals and reservoir hosts [16]. Alpaca herds have a close relationship with the human population, making MVA vaccination of SACs an important prophylactic tool because of the zoonotic potential of the disease. The duration of vaccinal immunity and the necessity for booster vaccinations after the initial immunisation need to be investigated in long-term studies. 

## Figures and Tables

**Figure 1 viruses-12-00234-f001:**
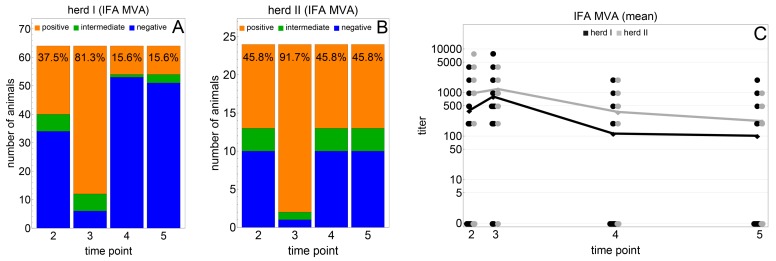
Field trial vaccination against cowpox in two alpaca herds in Germany: Results of indirect immunofluorescence assay (IFA) after immunisation with the MVA (modified vaccinia virus Ankara) vaccine in herd I (**A**) and herd II (**B**) and the distribution of MVA-specific IFA titres and mean titres by herd and time point (**C**). Herd I—Thuringia (*n* = 64); Herd II—Saxony Anhalt (*n* = 24); time point 2—four weeks after initial vaccination, time point 3—four weeks after booster vaccination, time point 4—six months after booster vaccination, time point 5—12 months after booster vaccination; solid dots/short horizontal lines—individual titres (Figure 1C); solid lines—mean titres (Figure 1C).

**Table 1 viruses-12-00234-t001:** Field trial vaccination against cowpox in two alpaca herds Germany: MVA vaccination regimen and sampling information.

Time Points	1 (Day Zero)	2 (+1 Mo)	3 (+1 Mo)	4 (+5 Mo)	5 (+6 Mo)
Vaccination	×	×	−	−	−
Adverse reaction	−	×	×	−	−
Clinical examination	×	×	×	×	×
Serum sample (antibody testing)	−	×	×	×	×
Swab sample (viral DNA)	−	×	×	×	×

× done/sampled; − not done/sampled; mo, months; MVA, modified vaccinia virus Ankara.

**Table 2 viruses-12-00234-t002:** Summary of individual antibody titres over time after prime-boost MVA vaccination against cowpox in alpacas.

Gr		1	2									3			4					
TP	P	1	2	3	4	5	6	7	8	9	10	11	12	13	14	15	16	17	18	total
**2**	**IFA (MVA)**	+	+	−	+	−	?	−	?	−	+	?	−	−	+	−	−	+	?	
**3**		+	+	+	+	+	+	+	+	+	−	?	?	−	+	+	+	+	+	
**4**		+	+	+	−	+	?	?	−	−	−	−	−	−	−	−	−	−	−	
**5**		+	?	+	−	−	−	−	−	−	−	−	−	−	+	+	?	?	?	
***n* herd I**		8	1		13	1		1	4	20	1	1	5	5	1	1	1		1	64
***n* herd II**		9	1	1			2	1	1	4			1	1		1	1	1		24
***n* total**		17	2	1	13	1	2	2	5	24	1	1	6	6	1	2	2	1	1	88

Gr, group; TP, time point (2—four weeks after initial vaccination, 3—four weeks after booster vaccination, 4—six months after booster vaccination, 5—12 months after booster vaccination); P, individual titre changes; *n*, number of animals; Herd: I—Thuringia; II—Saxony Anhalt; IFA, indirect immunofluorescence assay (+, positive; −, negative; ?, intermediate (>0<1:200, lowest dilution 1:200 tested reactive, but not positive)); MVA, modified vaccinia virus Ankara.

**Table 3 viruses-12-00234-t003:** Antibody titre changes using two different indirect immunofluorescence assays in the animals of group 3 (no seroconversion) and 4 (inconclusive titre changes).

Gr		3													4						
TP	P	11	12						13						14	15	15	16	16	17	18
**2**	**IFA (MVA)**	?	−	−	−	−	−	−	−	−	−	−	−	−	+	−	−	−	−	+	?
**3**		?	?	?	?	?	?	?	−	−	−	−	−	−	+	+	+	+	+	+	+
**4**		−	−	−	−	−	−	−	−	−	−	−	−	−	−	−	−	−	−	−	−
**5**		−	−	−	−	−	−	−	−	−	−	−	−	−	+	+	+	?	?	?	?
	**ID**	149	24	104	115	121	140	153	23	126	127	143	151	163	155	10	152	11	137	6	144
**2**	**IFA (CPXV)**	−	−	+	−	?	−	−	+	−	+	−	−	−	+	+	−	+	?	+	?
**3**		+	+	+	+	+	?	−	+	+	+	−	−	−	+	+	+	+	+	+	+
**4**		−	+	+	?	+	?	−	−	−	+	+	−	−	+	?	+	?	+	+	+
**5**		−	?	−	?	+	−	?	−	?	?	−	−	−	+	+	+	−	+	+	+

IFA, indirect immunofluorescence assay (+, positive; −, negative; ?, intermediate (>0<1:200, lowest dilution 1:200 tested reactive, but not positive)); MVA, modified vaccinia virus Ankara; CPXV, cowpox virus; TP, time point (2—four weeks after initial vaccination, 3—four weeks after booster vaccination, 4—six months after booster vaccination, 5—12 months after booster vaccination); Gr, group; P, individual titre changes; ID, animal identity number.

**Table 4 viruses-12-00234-t004:** Results of antibody testing in 14 crias and their dams as part of a field trial of MVA vaccination of alpacas against cowpox in Germany.

ID	HERD	SEX	AGE *	IFA Initial Sampling	MVA-VAC	IFA TP 5	Titre MARE **	MARE INTERVAL ***
**4**	II	m	3.3	1:500	+	−	+	<6 mo
**5**	II	f	3.7	1:500	+	−	+	<6 mo
**28**	II	f	5.1	1:200	+	−	+	<6 mo
**29**	II	f	3	<1:200	+	−	−	<6 mo
**30**	II	m	2.9	1:1,000	+	−	+	<6 mo
**31**	II	f	6.6	>0<1:200	+	−	+	<6 mo
**38°**	II	m	6.6	1:200	−	+	+	>6 mo
**39°**	II	m	12.3	<1:200	−	−	+	>6 mo
**40°**	II	f	7.3	>0<1:200	−	?	+	>6 mo
**168**	I	m	7.6	<1:200	−	−	−	>6 mo
**169**	I	m	3.7	<1:200	−	−	−	>6 mo
**170**	I	f	5.1	<1:200	−	−	−	>6 mo
**171**	I	f	3.6	1:500	−	−	+	>6 mo
**172**	I	f	4.3	1:1000	−	+	+	>6 mo

ID, animal identity number; Herd: I—Thuringia (*n* = 5); II—Saxony Anhalt (*n* = 9); m, male; f, female; IFA, indirect immunofluorescence assay (+, positive; −, negative; ?, intermediate or as endpoint dilution: ≥1:200, positive; <1:200, negative; >0<1:200, intermediate (lowest dilution 1:200 tested reactive, but not positive)); MVA-VAC, MVA vaccination (+, done; − not done); * age of the cria in weeks at time of initial sampling; ** antibody titre status of the mare at the time of birth; *** time interval between birth of the cria and latest MVA vaccination of the foaling mare (mo, months); ° time point initial sampling = time point (TP) 5 (Table 1).

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
