# Peer review of "Field Trial Vaccination against Cowpox in Two Alpaca Herds"

_viruses, 2020, doi:10.3390/v12020234_

Round 1

Reviewer 1 Report

This is a well written manuscript demonstrating the feasibility of safely vaccinating alpaca herds to protect them against spurious cowpox virus infection when in a European setting. The authors have used an immunofluorescence assay to evaluate serological immune responses in the animals.

The assay appears to be user dependent since it requires observation under the microscope. It would have been interesting if the authors had also been able to perform an assay that enables quantification with an instrument (quantitative reading using fluorescence microscopy or ELISA or a number of other methods).

It might also have been interesting to know whether methanol/ acetone fixation reduces to any extent the IFA assay as compared to a milder fixation.

It’s mentioned in the methods section that “Generally, sera scored positive only after viral plaque-associated staining”. This sentence implies that the authors observed viral plaques instead of cultures with confluent infection. The latter would be preferable to enable detection of fluorescence anywhere in the sample.

The sentence on page 4 line 180 reading “sera reactive, but not positive with titres between 0 and 200 were considered intermediate” would be clearer as follows “sera reactive with titres between 0 and 200 were considered intermediate”

For more clarity, one would like to know what was the lowest dilution of serum used? This would provide a better understanding of the basis for choosing the 1/200 dilution as the cutoff between positive and negative. Were the sera actually also used without any dilution?

Author Response

Response to Reviewer 1 Comments

Point 1: The assay appears to be user dependent since it requires observation under the microscope. It would have been interesting if the authors had also been able to perform an assay that enables quantification with an instrument (quantitative reading using fluorescence microscopy or ELISA or a number of other methods).

Response 1: We thank the reviewer for the helpful suggestions. Automated analysis would clearly contribute to objective and speedy evaluation of serum samples. However, we do not have established assays in the laboratory. Commercial products for serological analyses of animal sera are unfortunately not available either.

Point 2: It might also have been interesting to know whether methanol/ acetone fixation reduces to any extent the IFA assay as compared to a milder fixation.

Response 2: Interestingly, the indirect immunofluorescence assay used did not suffer from low-level fluorescence, but occasionally suffered from high background fluorescence. These individual animal sera exhibited non-specific staining of the cultured cells. That is why we scored only plaque-associated staining positive to rule out false positive evaluation. To further combat false positive results, sera were generally tested using 1:200 dilutions. The evaluation of paraformaldehyde fixation and digitonin permeabilisation did not result in different antibody levels in a study on bovine sera. We expect the same result with alpaca sera.

Point 3: It’s mentioned in the methods section that “Generally, sera scored positive only after viral plaque-associated staining”. This sentence implies that the authors observed viral plaques instead of cultures with confluent infection. The latter would be preferable to enable detection of fluorescence anywhere in the sample.

Response 3: As mentioned earlier we rely on plaque-associated staining to account for non-specific staining reactions. In addition, confluent infection of cowpox virus in cultured cells results in cytopathogenic effects that detach cells.

Point 4: The sentence on page 4 line 180 reading “sera reactive, but not positive with titres between 0 and 200 were considered intermediate” would be clearer as follows “sera reactive with titres between 0 and 200 were considered intermediate”

Response 4: The sentence was modified according to the reviewer’s suggestion. In the current version it is on page 4 line 178.

Point 5: For more clarity, one would like to know what was the lowest dilution of serum used? This would provide a better understanding of the basis for choosing the 1/200 dilution as the cutoff between positive and negative. Were the sera actually also used without any dilution?

Response 5: As mentioned earlier, all sera were diluted no less than 1/200. This procedure reduces non-specific background staining, and positive reactions are prominent. Some sera of alpacas were evaluated by plaque neutralization assay; however, this assay is much more time consuming, and live virus is used. To reduce the potentially hazardous exposure to a zoonotic agent and because the results of the plaque assay were in agreement with the results of immunofluorescence, we used IFA only for all sera.

Reviewer 2 Report

The Authors present a study on a prime-boost vaccination regimen in two alpaca herds with a history of CPXV infection. The modified vaccinia virus Ankara (MVA) vaccine used was well tolerated and safe in the 94 vaccinated animals. The authors performed an indirect immunofluorescence assay (IFA) using MVA as antigen showing that the seroprevalence of antibody after booster vaccination was high in both herds. According to this study, MVA vaccination seems a feasible tool for prevention of CPXV disease in alpacas and it is to be recommended every two to three years as described in elephants.

I have only the following minor comment to further strengthen the manuscript:

- line 134 – it would be helpful to the reader if you write on which site the animals received the the booster vaccination

Broadly, the study is well written, discussion is interesting, and the literature contains current contents.

Author Response

Response to Reviewer 2 Comments

Point 1: Extensive editing of English language and style required 

Response 1: English language and style were checked by a Native English Speaker before submission and meet the current requirements.

Point 2: line 134 – it would be helpful to the reader if you write on which site the animals received the booster vaccination

Response 2: We thank the reviewer for the helpful suggestion. Alpacas in our study were vaccinated twice at the same injection site – subcutaneously on the right side of the neck, cranial to the scapula. We added the requested information to the text on page 3 line 133.

Reviewer 3 Report

Prkno et al describe vaccination of alpacas with MVA to prevent cowpox virus (CPXV) disease. The authors took on the experiment because CPXV is responsible for zoonotic outbreaks in elephants, rhinoceri and other special populations. Transmission of CPXV from its reservoir to other species that can transmit the virus to humans has been reported.  

The experiment is well-defined to determine if MVA is effective against CPXV in camelids.  

The manuscript is well written, concise, and provides a new use for MVA to prevent disease in a special population.

Line 296 is "dams" appropriate or should it be mares?

Introduction and Discussion. Given the lack of data on cellular immunity the authors should qualify their recommendation about vaccination every 5 years.  Given that most alpacas did not show reactions to the vaccine it is likely safe, but the authors haven't demonstrated that the boost is necessary.

Author Response

Response to Reviewer 3 Comments

Point 1: Line 296 is "dams" appropriate or should it be mares?

Response 1: We thank the reviewer for the suggestion. We have used the expression “born to dams with positive….”, which we find correct and appropriate. The sentence was modified on page 8 lines 293-296.

Point 2: Introduction and Discussion. Given the lack of data on cellular immunity the authors should qualify their recommendation about vaccination every 5 years. Given that most alpacas did not show reactions to the vaccine it is likely safe, but the authors haven't demonstrated that the boost is necessary

Response 2: Thank you very much for this comment. We agree that the lack of data on cellular immunity as well as the lack of long-term studies on the duration of immune response after MVA vaccination do not allow statements on future vaccination regimen. We decided to remove this recommendation from the text (sections abstract and conclusions).To point out that this issue should be addressed in future studies, we modified the conclusions as follows (page 10, lines 407-408): ‘The duration of vaccinal immunity and the necessity for booster vaccinations after the initial immunisation need to be investigated in long-term studies.’